# Stochastic cell-intrinsic stem cell decisions control colony growth in planarians

Tamar Frankovits[1], Prakash Varkey Cherian[1], Yarden Yesharim[1], Simon Dobler[1†], Omri Wurtzel[1,2]*

[1]The School of Neurobiology, Biochemistry & Biophysics, George S. Wise Faculty of Life Sciences,Tel Aviv University, Tel Aviv, Israel; [2]Sagol School of Neuroscience, Tel Aviv University, Tel Aviv, Israel

*For correspondence:
owurtzel@tauex.tau.ac.il

Present address: †Interfaculty Institute of Microbiology and Infection Medicine, University of Tübingen, Tübingen, Germany

Competing interest: The authors declare that no competing interests exist.

## eLife Assessment

This manuscript establishes a mathematical model to estimate the key parameters that control the repopulation of planarian stem cells after sublethal irradiation as they undergo fate-switching as part of their differentiation and self-renewal process. The findings are **important** for future investigation of stem cell division in planarians and have implications for analyzing stem cell biology in other systems. The methods are **convincing**, integrating modeling with perturbations of key transcription factors known to be critical for cell fate decisions, but the authors have only shown that this is the case for a small number of stem cell types.

**Abstract** Stem cells contribute to organismal homeostasis by balancing division, self-renewal, and differentiation. Elucidating the strategies by which stem cells achieve this balance is critical for understanding homeostasis and for addressing pathogenesis associated with the disruption of this balance (e.g. cancer). Planarians, highly regenerative flatworms, use pluripotent stem cells called neoblasts to maintain and regrow organs. A single neoblast can rescue an entire animal depleted from stem cells and regenerate all cell lineages. How neoblast differentiation and clonal expansion are governed to produce all the required cell types remains unclear. Here, we integrated experimental and computational approaches to develop a quantitative model revealing basic principles of clonal growth of individual neoblasts. By experimentally suppressing differentiation to major lineages, we elucidated the interplay between colony growth and lineage decisions. Our findings suggest that neoblasts select their progenitor lineage based on a cell-intrinsic fate distribution. Arresting differentiation into specific lineages disrupts neoblast proliferative capacity without inducing compensatory expression of other lineages. Our analysis of neoblast colonies is consistent with a cell-intrinsic decision model that can operate without memory or communication between neoblasts. This simple cell fate decision process breaks down in homeostasis, likely because of the activity of feedback mechanisms. Our findings uncover essential principles of stem cell regulation in planarians, which are distinct from those observed in many vertebrate models. These mechanisms enable robust production of diverse cell types and facilitate regeneration of missing tissues.

## Introduction

Stem cells are crucial for achieving and maintaining homeostasis. Paradoxically, they contribute to an organism's equilibrium through highly dynamic behavior, continuously balancing their population size by regulating rates of self-renewal and differentiation. Understanding the strategies used to accomplish this balance can provide the key to elucidating organismal homeostasis, as well as to modeling

pathogenesis associated with disruption to this balance (e.g. cancer) (*Passegué et al., 2003*; *Singh et al., 2011*).

Valuable insights into stem cell behavior can be derived from studying systems that rely on principles differing significantly from those characterizing vertebrate stem cells. Neoblasts, the pluripotent stem cells in planarians, function in tissue regeneration and maintenance, providing an intriguing platform in this regard. Neoblasts avoid quiescence and quickly adapt to changes in organismal requirements by adjusting the quantities and types of cells they produce (*González-Estévez et al., 2012*; *Wagner et al., 2011*). These strategies challenge fundamental concepts of vertebrate stem cells. For example, in adult vertebrates, hematopoietic stem cells typically divide slowly, giving rise to rapidly dividing transit amplifying cells forming tissue-specific progenitors (*Abkowitz et al., 2000*; *Becker et al., 2006*). These progenitors bear the primary responsibility for maintaining homeostasis. Planarian neoblasts, by contrast, divide rapidly and do not employ transit amplifying cells for progeny production (*Pearson, 2022*; *Raz et al., 2021*; *van Wolfswinkel et al., 2014*). Moreover, neoblasts do not seem to retain substantial memory of the progeny cell types they generate (*Raz et al., 2021*; *van Wolfswinkel et al., 2014*). Notably, despite being essentially ageless and immortal, neoblasts do not display cancerous-like states (*Van Roten et al., 2018*). Therefore, elucidating how neoblasts balance self-renewal and differentiation can contribute to understanding fundamental strategies that facilitate tissue homeostasis and regeneration of lost or damaged tissues (*Levin et al., 2019*).

Prior modeling of neoblast dynamics in planarians has produced a valuable analytical framework for studying the dynamics of the entire neoblast system (*Mangel et al., 2016*). However, analyzing isolated neoblasts and their progeny can be instrumental for understanding how decisions are made at a single-cell resolution, as well as for deriving physiological parameter values (e.g. corresponding to proliferation and differentiation) that are relevant to the larger system. *In vivo* analysis of individual neoblasts and their progeny is often achieved by applying subtotal irradiation to whole planarians, which results in near-complete neoblast ablation (*Wagner et al., 2011*). Surviving neoblasts proliferate to form distinct colonies, allowing analysis of growth dynamics and progeny identity (*Lei et al., 2016*; *Wagner et al., 2012*). Though such analysis has been instrumental in uncovering regulators

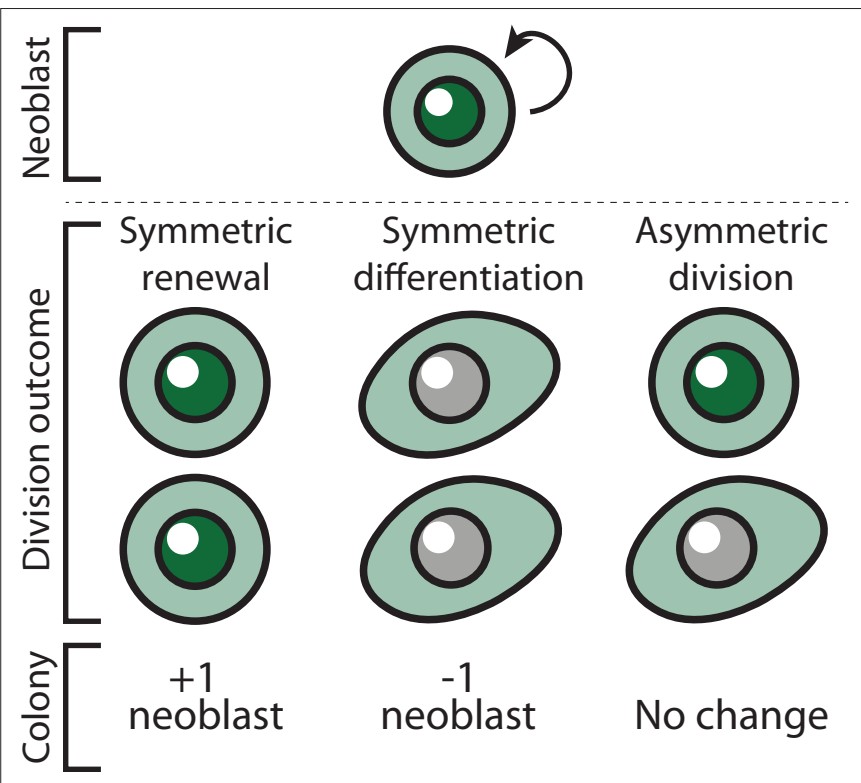

**Figure 1.** Potential outcomes of neoblast division and the effect on colony size.

of proliferation and differentiation (*Wagner et al., 2012*), a quantitative analysis of neoblast colony growth that integrates these principles has yet to be established and experimentally validated.

Several open questions need to be addressed in the development of such a model. For example, it is unclear whether neoblasts select their lineage identity in a synchronous manner or function independently (*Bohr et al., 2021*; *Reddien, 2018*). Moreover, as indicated above, it is unclear whether neoblasts maintain memory of their decisions over successive rounds of the cell cycle (*Pearson, 2022*). In addition, as discussed in what follows, critical knowledge is lacking regarding the ability of neoblasts to switch from producing differentiating lineage to contributing to neoblast colony growth; an understanding of this switch is fundamental for analyzing the overall development of a colony, and for describing new tissue production in regeneration.

Herein, we begin to address these gaps, integrating experimental and computational approaches to analyze neoblast colony growth. Our analysis does not assume communication between neoblasts or memory of cell fate decisions; rather, it explores whether a straightforward stochastic selection of division outcome can adequately describe the growth of a neoblast colony. Based on *in vivo* analysis of neoblast clone growth, neoblast division can result in three potential outcomes (*Figure 1*; *Lei et al., 2016*; *Raz et al., 2021*): (1) symmetric renewal, which generates two neoblasts; (2) symmetric differentiation, which generates two post-mitotic cells; and (3) asymmetric division, which produces a neoblast and a post-mitotic cell.

In what follows, we study and analyze colony growth based on these principles. We derive key parameter values from experimental data and show that these values predict colony growth with high accuracy. Next, in a series of experiments, we seek to uncover whether the core principles that drive neoblast colony growth (as proposed in our model) are further shaped by individual neoblasts' decisions regarding progeny lineage commitment. In particular, many S/G2/M neoblasts express fate-specifying transcription factors (FSTFs), which direct the differentiation of their progeny into specific lineages (*Reddien, 2013*; *Scimone et al., 2014a*; *Zeng et al., 2018*). Importantly, a neoblast can express different FSTFs in successive rounds of division, which results in the production of post-mitotic progenitors of different lineages (*Raz et al., 2021*; *van Wolfswinkel et al., 2014*). Thus, the growth potential of the colony is not compromised by FSTF expression. Yet, it remains unclear how the growth of a colony might be affected when the expression of a particular FSTF (and thus the corresponding cell lineage) is blocked: e.g., does colony growth remain stable, with other lineages being overexpressed to compensate for the blocked lineage? Notably, our experimental perturbation analysis shows that blocking the expression of an FSTF incapacitates a proportion of neoblasts equivalent to the size of the lineage affected. This observation reveals a critical dependency: once a neoblast commits to a specific lineage, it retains this identity until cell division concludes. By contrast, inhibiting FSTFs during homeostasis resulted in an increase in the number of cycling cells, consistent with an activity of feedback mechanisms that regulate cell proliferation in homeostasis (*Cheng et al., 2018*; *Pearson and Sánchez Alvarado, 2010*).

Together, our findings imply that simple design principles govern cell fate choices in neoblast colonies, and that accurate prediction of colony growth does not require assumptions of fate specification memory or coordination of cell fate choices between neoblasts. Moreover, our findings suggest that all neoblasts in a colony are functionally equivalent (i.e. able to self-renew and to divide to all cell types) and elucidate the contribution of decisions in individual planarian stem cells to achieving systemic homeostasis.

## Results
### Analysis of neoblast colony growth
Following subtotal irradiation, a surviving neoblast may proliferate to form a colony (*Figure 2A*). The growing colony is the exclusive source of new cells (*Wagner et al., 2012*; *Wagner et al., 2011*). We used an exponential growth equation to estimate the change in colony size over time of a successfully established colony by a surviving neoblast (*Equation 1*). The function incorporates three parameters to estimate colony growth: (1) average cell cycle length ($\tau$); (2) symmetric renewal probability ($p$), where symmetric renewal rate per hour is $p'=p/\tau$; and (3) symmetric differentiation or neoblast elimination probability ($q$), where symmetric differentiation or neoblast elimination rate per hour is $q'=q/\tau$. Asymmetric divisions do not alter neoblast count (*Figure 1*).

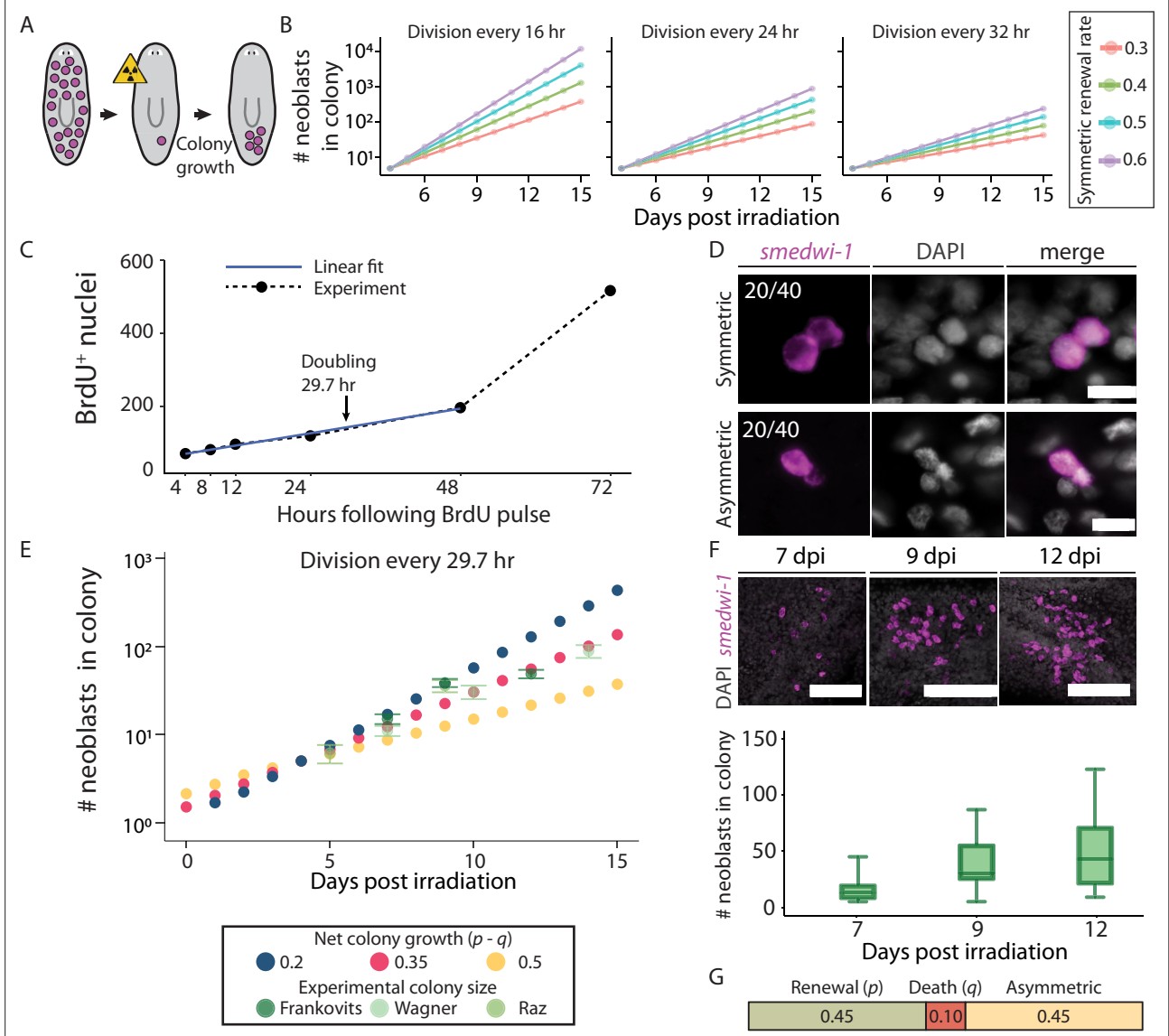

**Figure 2.** Parameters determining neoblast colony size. (**A**) Outline of colony growth following subtotal irradiation. Planarians are irradiated to eliminate all neoblasts (purple dots) except for a single survivor. The surviving neoblast forms a proliferating neoblast colony. (**B**) The impact of average cell cycle length ($\tau$) and probability of symmetric renewal division (*p*) on colony size is shown (*q* was set to 0). (**C**) Reanalysis of BrdU incorporation time series (***Lei et al., 2016***) in expanding planarian colonies was used for determining the average cell cycle length (Methods). Linear regression (blue line; y=59.77 + 2.81 * t; Pearson r=0.997) was used to identify the doubling time of the colony, and was determined as 29.7 hr (black arrow). (**D**) The fraction of symmetric renewal divisions was estimated by examining neoblast pairs in colonies at 7 days post-irradiation (dpi) (Methods). Scale = 10 µm. (**E**) The impact of symmetric differentiation or neoblast elimination (*q*) on colony size was tested, assuming a 29.7 hr long cell cycle. Experimental data of colony sizes at different time points were overlaid to determine the value of biologically relevant parameters. Frankovits, data collected here; ***Wagner et al., 2012***; Raz, reanalysis (***Raz et al., 2021***). Error bars show the standard error. (**F**) Shown are neoblast colonies at three time points following subtotal irradiation. Representative images (z-projection) are shown (top), and neoblast counting of different colonies is shown (bottom; box indicates interquartile range [IQR]; whiskers show range; horizontal bar indicates median). Scale = 100 µm. (**G**) Summary of the estimated probabilities of divisions leading to symmetric renewal, symmetric differentiation or elimination, and asymmetric divisions.

The online version of this article includes the following figure supplement(s) for figure 2:

**Figure supplement 1.** Analysis of colony growth.

$$\frac{dN}{dt} = p^{'}\text{N} - q^{'}\text{N} = \frac{(p-q)\,\text{N}}{\tau} \tag{1}$$

We used the solution to the equation and included the initial colony size ($N_0$) to predict colony sizes (*Equation 2*) using a range of parameter values and evaluated the model predictions (*Figure 2—figure supplement 1*) with our own and published data (*Raz et al., 2021*; *Wagner et al., 2012*).

$$N\left(t\right) = \text{N}_0\text{e}^{\dfrac{(p-q)\,\text{t}}{\tau}} \tag{2}$$

The plots in *Figure 2B* show how varying the value of growth parameters is predicted to affect colony size. The average cell division time ($\tau$) has the most substantial impact on colony growth. For example, a simulated colony with a fast (16 hr) cell cycle (with *p* and *q* set at 0.5 and 0, respectively) had a neoblast count approximately 10 times larger than that reported in literature (*Lei et al., 2016*; *Raz et al., 2021*; *Wagner et al., 2012*; *Wagner et al., 2011*; *Zeng et al., 2018*). To determine the average cell cycle time ($\tau$), we reanalyzed available colony growth data that had previously been experimentally obtained using bromodeoxyuridine (BrdU) metabolic labeling (*Lei et al., 2016*). In those experiments, BrdU was incorporated into DNA during both symmetric and asymmetric divisions using a pulse (4 hr) significantly shorter than the cell cycle time, followed by a chase of up to 72 hr (*Figure 2C*). We predicted that the BrdU⁺ cell number would grow linearly until the entire population doubled. Then, completion of successive cell divisions would lead to exponential growth in BrdU⁺ cell number. Indeed, the colony growth was initially linear (Pearson r=0.997), and using linear regression, we estimated that the colony doubling time, and hence the average cell cycle length, was 29.7 hr (*Figure 2C*).

The symmetric renewal probability (*p*) is a vital factor in colony growth, and even slight variations in *p* have large cumulative effects (*Figure 2B*). We experimentally determined the symmetric renewal probability by subjecting planarians to subtotal irradiation (1750 rad; Methods) and identifying neoblasts via fluorescence in situ hybridization (FISH) with the *smedwi-1* marker (*Reddien et al., 2005b*), a pan-neoblast marker. We imaged pairs of adjacent *smedwi-1*⁺ cells in the colony and classified the *smedwi-1*⁺ cell pairs based on their *smedwi-1* expression levels: high-level pairs indicated symmetric renewal, while pairs with one high and one low *smedwi-1*⁺ cell were categorized as asymmetric division (*Figure 2D*). By analyzing 40 dividing pairs, we estimated that 50% of divisions that produce at least one neoblast are symmetric renewal divisions, in agreement with previous findings (*Lei et al., 2016*; *Raz et al., 2021*).

The value of q, representing symmetric differentiation and neoblast loss probability (*Raz et al., 2021*), also plays a role in colony dynamics (*Figure 2E*, *Figure 2—figure supplement 1B*). Yet the value of this parameter is more difficult to estimate experimentally (e.g. naïve measurement of cell death using TUNEL in irradiated planarians would be insufficient, because cell death is not limited to neoblasts). We analyzed the effect of different neoblast elimination probabilities on colony size (*Figure 2E*, *Figure 2—figure supplement 1B–E*) and compared the estimates of the analysis with colony size in our experimental data and by extracting colony size measurements from published literature (*Figure 2E and F*; Methods; *Raz et al., 2021*; *Wagner et al., 2012*; *Wagner et al., 2011*). Considering an average cell cycle length of 29.7 hr, we calculated the value of *q* using the following approach: the probabilities of all cell division outcomes must sum to 1. Our experimental data showed that symmetric renewal (*p*) and asymmetric division (*a*) occur at equal rates (i.e. *p*=*a*). By fitting these parameters to the experimental data, we determined that the difference between the probabilities of symmetric renewal and symmetric differentiation (i.e. *p – q*) was = 0.345 (*Figure 2E*, *Figure 2—figure supplement 1D and E*). Therefore, with these criteria, we estimated the probabilities of cell division outcomes in the colony as *p*=0.45, *a*=0.45, and *q*=0.1 (*Figure 2G*; Methods). *In vivo*, we observed a large variation in the size of colonies at each time point (*Figure 2F*). This could be a consequence of an altered delay between irradiation and the onset of colony growth, or from a stochastic decision to undergo symmetrical or asymmetrical division in the early colony. Despite this variability, the observed exponential growth suggested a steady proliferation rate once the colony founding neoblast recovered at day 4 (*Figure 2E*, *Figure 2—figure supplement 1A*).

## Inhibition of lineage differentiation reduces neoblast colony size

Regulating growth and homeostasis requires balancing self-renewal and progeny production. To better understand how neoblasts maintain this balance, we focused on the interplay between the overall growth of the colony and the ability of individual neoblasts to express FSTFs, which are required for specific progeny production (*Scimone et al., 2014a*). More specifically, we investigated how inhibiting differentiation into specific lineages, by suppressing FSTF expression, influences colony size. Considering the linear correlation between post-mitotic progeny production and neoblast number (*van Wolfswinkel et al., 2014*; *Wagner et al., 2012*), we hypothesized three potential outcomes (*Figure 3A*) and their probable effects on symmetric renewal (*p*) and symmetric differentiation or elimination (*q*): (1) Unchanged: *p* and *q* remain balanced, resulting in no change in colony size. This outcome would suggest that the balance of self-renewal and progeny generation is preserved through compensatory production of alternative cell types when differentiation to a specific lineage is suppressed. (2) Larger: An increase in *p*, leading to larger colonies, possibly due to fewer asymmetric divisions in the blocked lineage. (3) Smaller: An increase in *q*, leading to smaller colonies, possibly because neoblasts committed to the blocked lineage have become dysfunctional. We used published frequencies of three major lineages (epidermis, intestine, and *foxF-1*+ neoblasts) to estimate the size of the colony in each of the described scenarios (*Figure 3A*; *Raz et al., 2021*; *Scimone et al., 2014a*; *Scimone et al., 2018*; *van Wolfswinkel et al., 2014*). Theoretically, inhibiting a predominant lineage, such as the epidermal lineage (28%), may significantly affect colony size, which could be observed experimentally, whereas changes in colony size following inhibition of smaller lineages could be difficult to distinguish within the experimental timescale.

On the basis of this assumption, we inhibited differentiation into the epidermal lineage after subtotally irradiating planarians. Following a 4-day recovery after subtotal irradiation, we suppressed the FSTF *zfp-1*, which is critical for the production of differentiating epidermal progenitors (*Figure 3—figure supplement 1*; *Cheng et al., 2018*; *Tu et al., 2015*; *van Wolfswinkel et al., 2014*; *Wagner et al., 2011*), but does not impact directly other lineages (*van Wolfswinkel et al., 2014*). Then, we evaluated neoblast colony sizes in *zfp-1*-suppressed (i.e., *zfp-1* (RNAi)) planarians at three time points (7, 9, and 12 days post-irradiation [dpi]), and compared them with control colony sizes (*Figure 3B*, Methods). Initially (7 dpi), colony sizes were similar. However, at 9 dpi, *zfp-1* (RNAi) colonies were significantly smaller than in controls. Moreover, unlike in controls, *zfp-1* (RNAi) colonies showed no additional growth, on average, during the period between 9 dpi and 12 dpi, though there was a slight increase in the median colony size (*Figure 3B*).

Our basic assumption was that reduction in colony size (vs. control) following *zfp-1* inhibition would indicate an increase in symmetric differentiation or elimination (*q*). However, an alternative possibility is that *zfp-1* inhibition causes mitotic arrest in neoblasts – which would prevent proliferation but would not necessarily result in cell death. We tested whether *zfp-1* (RNAi) colonies had fewer cycling neoblasts. We detected neoblasts in mitosis using immunofluorescence (IF) with an anti-H3P-antibody (*Figure 3C*; Methods). We found that control colonies did not significantly differ from *zfp-1* (RNAi) colonies in the number of H3P+ cells, either in absolute numbers or normalized to average neoblast counts in colonies (Methods). This result suggested that FSTF inhibition did not cause a general mitotic arrest. Further analysis with 2'-deoxy-2'-fluoro-5-ethynyluridine (EdU) labeling (*Figure 3D*; Methods) showed a substantial decrease in EdU+ nuclei in *zfp-1* (RNAi) animals (vs. controls), indicating reduced new cell production. However, normalizing EdU+ nuclei to average colony size (*Figure 3D*; Methods) revealed a comparable proportion of cells entering S-phase. Given the unchanged ratios of cycling and mitotic cells, we can conclude that the diminished colony size following *zfp-1* suppression is indeed likely to result either from increased symmetric differentiation or neoblast elimination (*q*). We further suggest that neoblast elimination is a more plausible explanation than increased symmetric differentiation: If *zfp-1* suppression were to lead to increased symmetric differentiation, we would expect to observe a noticeable decline in the number of neoblasts within colonies, while proliferation rates remained stable. Yet, the nonsignificant difference in EdU-labeling (*Figure 3D*) between *zfp-1* (RNAi) and control colonies suggested that inhibition of the FSTF *zfp-1* likely led to increased neoblast elimination.

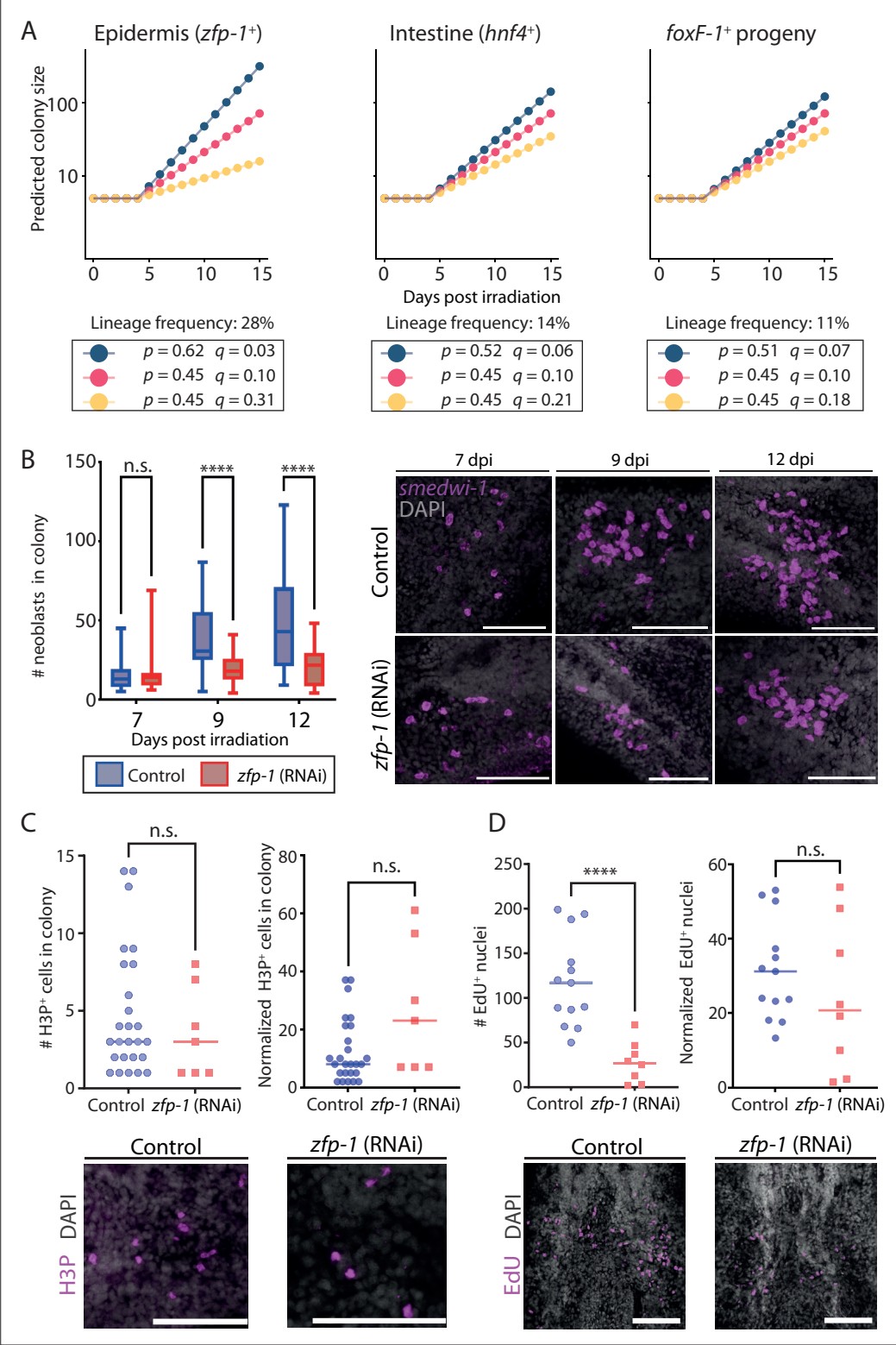

**Figure 3.** Reduced neoblast colony size following lineage specification block. (**A**) Models for lineage growth. Potential outcomes of lineage inhibitions were assessed by altering the colony growth and degradation parameters. Applying known lineage frequencies (*Raz et al., 2021*) to the models indicated that inhibiting production of a major lineage could dramatically impact colony size (left, epidermis). By contrast, inhibition of smaller lineages (middle, right) might have a smaller impact, which would be difficult to detect (blue: increase

*Figure 3 continued on next page*

*Figure 3 continued*

in symmetric renewal; red: unchanged renewal and degradation; yellow: increase in neoblast degradation). Growth was calculated using the following values: $N_0$=5, $\tau$ =29.7 hr, and growth lag time of 5 days (*Raz et al., 2021*; *Wagner et al., 2012*). (**B**) Shown are counts (left) of neoblasts in colonies at three time points following irradiation, which was followed by inhibition of *zfp-1* or control by RNAi. Lineage inhibition resulted in a highly significant decrease in colony size at later time points (box indicates interquartile range [IQR], whiskers show range, bar indicates the median). Number of control colonies analyzed at 7 days post-irradiation (dpi) n = 28; at 9 dpi n = 30; at 12 dpi n = 30; number of *zfp-1* (RNAi) colonies analyzed at 7 dpi n = 15; at 9 dpi n = 30; at 12 dpi n = 30. Representative colonies (z-projection) are shown (right). (**C**) Comparison of absolute (top-left) or normalized (top-right) H3P⁺ cell numbers in *zfp-1* (RNAi) and control colonies at 12 dpi showed a nonsignificant difference (Methods). Representative H3P labeling images are shown (bottom). Importantly, the number of detectable H3P⁺ cells was small. (**D**) Comparison of EdU⁺ nuclei (top-left) in *zfp-1* (RNAi) and control colonies at 12 dpi showed a significant reduction, contributing to the smaller colony size. Comparison of normalized EdU⁺ nuclei numbers (top-right) showed a nonsignificant difference, indicating that a similar proportion of neoblasts in the colony were cycling (Methods). Representative EdU labeling images (z-projection) are shown (bottom). Statistical significance was assessed using the Mann-Whitney two-tailed test (Methods). n.s., not significant, **** p<0.0001. Scale = 100 µm.

The online version of this article includes the following figure supplement(s) for figure 3:

**Figure supplement 1.** Inhibition of epidermal progenitors by a single *zfp-1* dsRNA injection.

## Specialized neoblasts are produced in *zfp-1* (RNAi) colonies

Building upon our findings regarding lineage-specific inhibition effects, we next investigated how RNAi of *zfp-1* affected the prevalence of other specialized neoblasts in colonies. We hypothesized that if neoblast elimination was the primary effect of *zfp-1* inhibition, we would observe an absolute decrease in specialized neoblasts of other lineages (e.g. intestine), but not necessarily a relative decrease. We used subtotal irradiation, and following recovery, we inhibited *zfp-1*. Then, we counted specialized neoblasts expressing intestine lineage markers (*Scimone et al., 2014a*; *Wagner et al., 2011*) and *tgs-1* (*Raz et al., 2021*; *Zeng et al., 2018*), which is suggested to label neural progenitors (*Figure 4A and B*; Methods). Both control and *zfp-1* (RNAi) colonies contained specialized neoblasts. Notably, a decrease in intestine specialized neoblasts was detectable, aligned with the reduced colony size (*Figure 4A*). No change was observed in *tgs-1*⁺ cells, likely due to the low number of *tgs-1*⁺ cells detected in the colony (two cells on average) (*Figure 4B*). Together, these results suggested that specialized neoblasts were produced in *zfp-1* (RNAi) colonies, but that production of other (non-suppressed) lineages did not increase. This indicated that neoblast elimination predominantly accounted for the reduction in colony size following suppression of the epidermal lineage, without compensation by overproduction of other lineages.

To further analyze the impact of *zfp-1* inhibition on colony growth, we examined the numbers of neoblasts in 64 control colonies and in 53 *zfp-1* (RNAi) colonies, at 12 dpi (*Figure 4C and D*). The proportion of colonies containing five or more neoblasts was greater in the control group (75%) than in the *zfp-1* (RNAi) group (45%; Fisher's exact test two-tailed p=2.4 × 10⁻⁵). Importantly, more *zfp-1* (RNAi) animals failed to develop colonies compared to controls (55% and 25%, *zfp-1* (RNAi) and control colonies, respectively). This might have resulted from the stochastic selection of a *zfp-1* identity by the neoblasts that initially established the colony, which led to their further dysfunction, and was therefore detrimental to the development of the colony.

## Simulation of colony growth following *zfp-1* inhibition

We developed a simulation to recapitulate the observed reduction in colony growth following *zfp*-1 inhibition. Our simulation methodology was designed to mimic real-world neoblast dynamics under *zfp-1* inhibition in a colony, using empirically derived probabilities for each cell cycle outcome. First, we randomly sampled initial colony sizes from the set counted at 7 dpi (*Figure 3B*). In every simulation cycle, each neoblast fate was selected based on the probabilities of cell cycle outcomes identified above, derived from empirical data (symmetric renewal, asymmetric division, symmetric differentiation, or elimination; *Figure 2G*). The change in colony size was calculated over approximately six cell cycles. We conducted 100 iterations per simulation and compared the results with experimental data (*Figure 4E*; Methods). The simulated colony sizes closely matched the experimental observations.

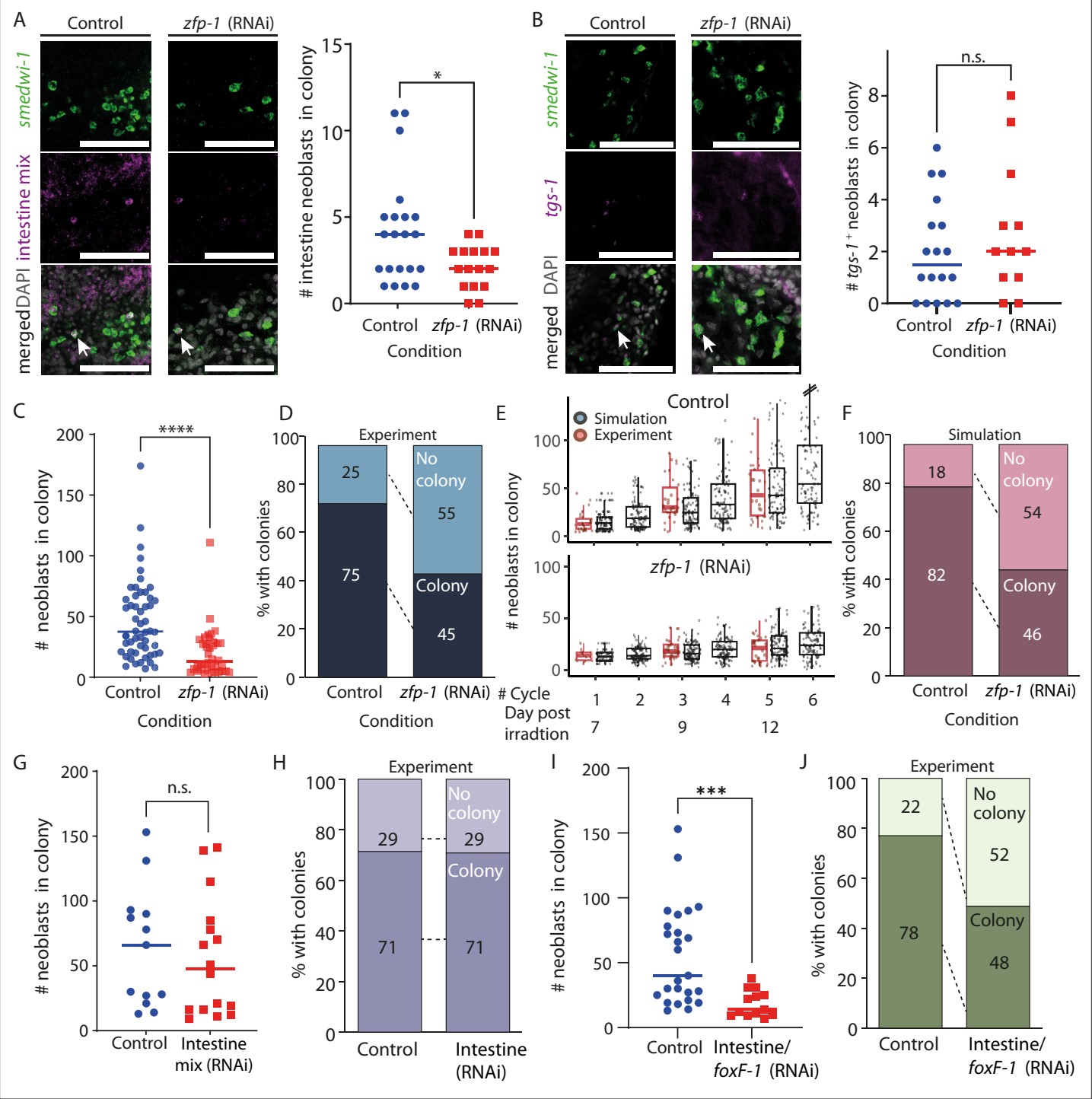

**Figure 4.** Simulation of neoblast colony growth following lineage block. (**A, B**) Neoblasts expressing lineage gene expression markers were detected using fluorescence in situ hybridization (FISH) (intestine mix: *hnf-4, gata4/5/6,* and *nkx2.2*; *tgs-1*) and were counted in colonies of control and *zfp-1* (RNAi) animals at 12 days post-irradiation (dpi) (Methods). Left panels show confocal images of representative colonies (Methods); quantification of the experiment is shown on the right. Scale = 100 µm. (**C**) Comparison of colony size in control and in *zfp-1* (RNAi) animals 12 dpi showed a highly significant reduction in the size of *zfp-1* (RNAi) colonies (Mann-Whitney two-tailed test, **** p<0.0001). Data is also shown in *Figure 3B*. (**D**) Summary of colony production in control and *zfp-1* (RNAi) animals based on analysis of 64 and 53 control and *zfp-1* (RNAi) colonies, respectively, at 12 dpi. Developed: ≥5 neoblasts detected; no colony <5 neoblasts detected. (**E**) Simulation of colony growth in control and *zfp-1* (RNAi) animals, bottom and top, respectively (Methods). Shown are multiple iterations (black dots; n=100 for each condition) of colony size simulation starting at 7 dpi. Experimental data quantifying colony sizes are shown (red dots) next to the corresponding simulation cycle. Box indicates the interquartile range (IQR); whiskers indicate ±1.5 × IQR;

*Figure 4 continued on next page*

*Figure 4 continued*

values out of the whisker range (outliers) were removed for clarity. (**F**) Summary of colony production in simulated control and *zfp-1* (RNAi) animals based on experimentally determined lineage frequencies, and assuming no memory in division outcome decision. Developed colony: ≥5 neoblasts in the simulation on day 12. (**G**) Neoblasts were counted in colonies at 12 dpi following the inhibition of intestine progenitors by combined RNAi treatment (*hnf-4*, *gata4/5/6*, *nkx2.2*). Lineage inhibition did not affect colony size (Mann-Whitney two-tailed test, p=0.404; n=13 and 16, control and RNAi animals, respectively). (**H**) Suppression of intestine lineage did not affect the likelihood of producing colonies (n=21 and 24, control and RNAi animals, respectively). (**I, J**) Combined suppression of *foxF-1* (***Scimone et al., 2018***) and intestine lineage production resulted in smaller colonies (Mann-Whitney two-tailed test, p=3 × 10$^{-4}$) and a significant reduction in the likelihood of producing colonies (Fisher's exact test two-tailed p=1.8 × 10$^{-5}$).

The online version of this article includes the following figure supplement(s) for figure 4:

**Figure supplement 1.** Simulation of colony size following RNAi.

When simulating *zfp-1* (RNAi) colony growth by increasing the symmetric differentiation or elimination probability (*q*) by the known proportion of *zfp-1*$^+$ neoblasts, the model recapitulated the experimental findings (***Figure 4E and F***). Notably, many simulated *zfp-1* (RNAi) colonies failed to grow, often because of initial sampling of a state of symmetric differentiation or elimination, prohibiting further development of the colony (***Figure 4F***).

The agreement of these simulations with the experimental results is aligned with a model where neoblasts independently select their cell cycle outcome based on a replication outcome distribution without maintaining any memory of previous replication outcomes. Moreover, inhibition of a gene required for neoblast differentiation (e.g. FSTF) impedes the contribution of neoblasts that randomly selected this identity from any further contribution to colony growth across the timescale of the simulation.

## Combined suppression of low-frequency lineages affects neoblast colony formation

Our model and simulations indicate that inhibition of smaller lineages will not result in detectably smaller colonies (***Figure 4—figure supplement 1A–C***). We tested this model prediction by comparing colonies produced following suppression of intestine lineage progenitors (***Forsthoefel et al., 2012***; ***van Wolfswinkel et al., 2014***; ***Wagner et al., 2011***), which account for 14% of the produced progenitors (***Raz et al., 2021***). Indeed, highly efficient inhibition of the intestine lineage by RNAi (***Figure 4—figure supplement 1D***) did not result in reduced colony size or colony production, in agreement with our model (***Figure 4G and H***, ***Figure 4—figure supplement 1A***). This result corroborated the model's predictions regarding lower-frequency lineages (***Figure 3A***): the potential increase in *q* following suppression of differentiation to a lower-frequency lineage had an undetectable effect on colony size. Furthermore, the model predicts that inhibition of several lineages, which collectively amount to a larger fraction of produced progenitors, may generate a detectable difference in colony size and successful colony establishment. We co-suppressed the production of intestine lineage together with *foxF-1*$^+$ progenitors, which together amount to ~25% of the progenitors (***Raz et al., 2021***), following subtotal irradiation (***Figure 4I and J***). The co-inhibition of the two lineages resulted in a highly significant decrease in colony size (***Figure 4I***) and a significant increase in the failure to establish colonies (***Figure 4J***; 12/23; 52%), compared to their controls (4/18; 22%; Fisher's exact test two-tailed p=1.8 × 10$^{-5}$).

These results reinforce the hypothesis that FSTF inhibition reduces colony growth (vs. control), primarily through increased neoblast elimination, without compensatory increases in other lineages. The results also suggest that a neoblast that has selected an identity, but that cannot complete proliferation and differentiation, will not be able to adopt any other identity in this experimental timescale. Based on these conclusions, we propose a semi-stochastic model for selecting the outcome of neoblast division, assuming no memory or coordination between cells, that is sufficient for describing neoblast colony growth.

## Inhibition of *zfp-1* does not induce overexpression of other lineages in homeostasis

Clonal analysis is a powerful approach for studying neoblast proliferation and progeny production. However, stem cell behavior in homeostasis may differ from that observed in clonal growth.

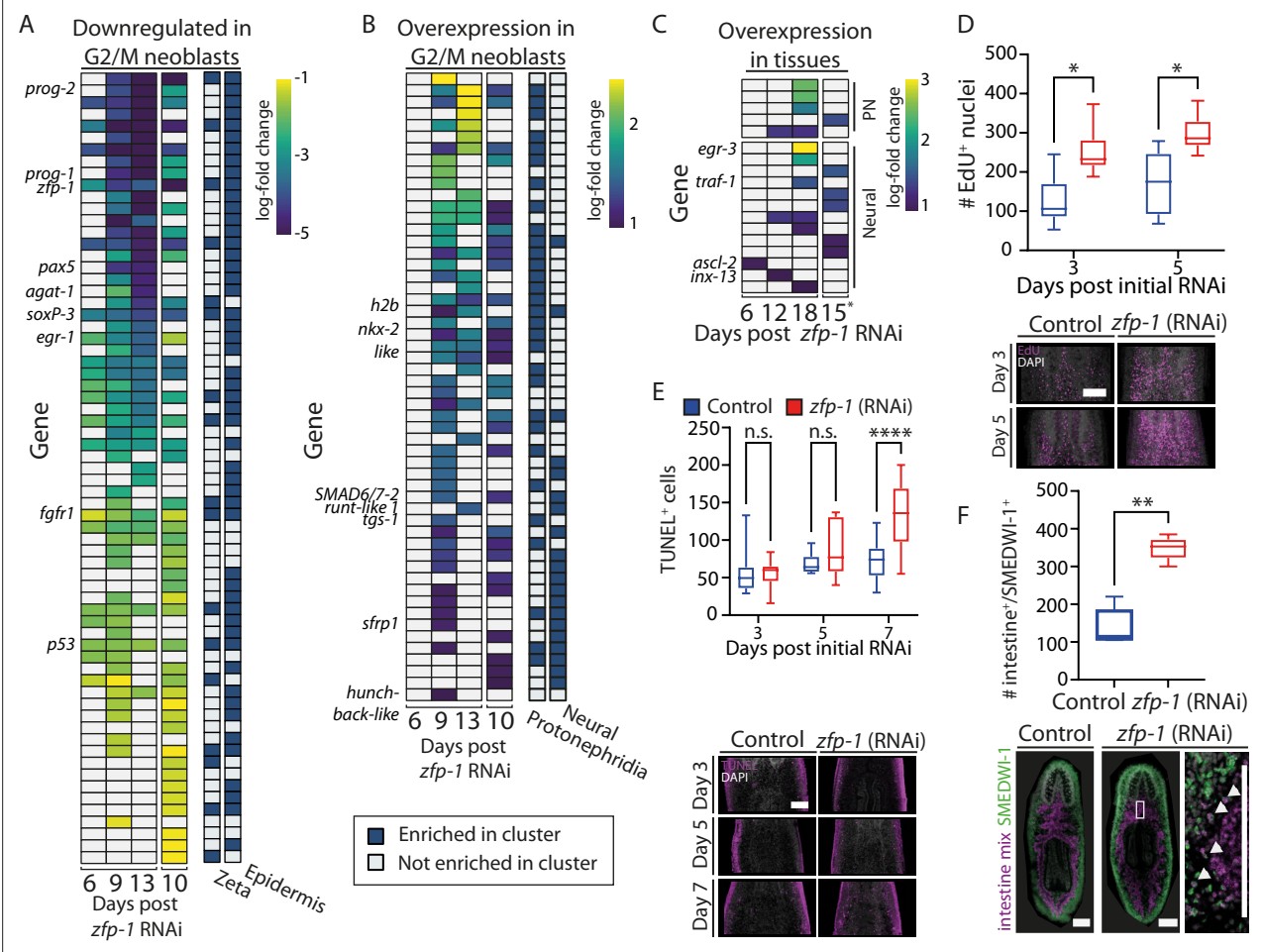

**Figure 5.** Analysis of *zfp-1* inhibition consequences in homeostasis. (**A**) Downregulated genes in FACS-purified S/G2/M neoblasts were overwhelmingly associated with expression in the epidermal lineage and epidermis-specialized (zeta) neoblasts (**Cheng et al., 2018**; **Fincher et al., 2018**; **van Wolfswinkel et al., 2014**). (**B**) Upregulated genes in FACS-purified S/G2/M neoblasts were largely associated with neuronal and protonephridial specialized gene expression (**Fincher et al., 2018**). (**C**) Analysis of upregulated genes of whole tissues following *zfp-1* inhibition showed that only a few factors were associated with protonephridia or neurons. (**A–C**) Heatmaps showing row-scaled gene expression obtained from PLANAtools (**Hoffman and Wurtzel, 2023**). Blue and yellow, low and high log-fold gene expression difference, respectively. Columns on the right indicate cell type-specific gene expression (**Fincher et al., 2018**). (**D**) Counting EdU+ nuclei in unirradiated control and *zfp-1* (RNAi) animals. (**E**) Counting apoptotic cells in unirradiated control and *zfp-1* (RNAi) animals (Methods). (**F**) Counting intestinal progenitors showed an increase in progenitors at later time points, in agreement with the gene expression analysis of *zfp-1* (RNAi) of whole tissues. Scale = 100 μm. * p<0.05; ** p<0.01; **** p<0.0001.

For example, unlike in clonal expansion, the stem cell population size remains unchanged during homeostasis, indicating the differential activity of feedback mechanisms. Moreover, the vastness of the neoblast population in homeostasis allows us to examine the ability of neoblasts to respecify their lineage over time and to determine if FSTF inhibition results in an increase in cell loss rate.

We examined whether the suppression of a major lineage production has resulted in overproduction of other lineages, a phenomenon that we did not observe in colonies. First, we used two published gene expression datasets from FACS-purified S/G2/M neoblasts in *zfp-1* (RNAi) and control animals collected at various time points (**Cheng et al., 2018**; **van Wolfswinkel et al., 2014**; Methods). We utilized pre-calculated tables of differentially expressed genes in these datasets (**Hoffman and Wurtzel, 2023**) and annotated the cell types that express them (**Fincher et al., 2018**). Most downregulated genes with known neoblast lineage enrichment (**Fincher et al., 2018**) were epidermal (82%, 55/67; **Figure 5A**; **Supplementary file 1**; Methods). Conversely, upregulated genes following *zfp-1* suppression were primarily associated with protonephridial or neural neoblasts (30/57 and 20/57 genes, respectively; **Figure 5B**; **Supplementary file 2**), consistent with the original data analysis (**Cheng et al., 2018**). Notable upregulated genes included transcription factors, such as *nkx2-like*

(*Currie et al., 2016*), *hunchback-like* (*Reddien et al., 2005a*), *runt-1* (*Sandmann et al., 2011*; *Wene-moser et al., 2012*), and *tgs-1* (*Zeng et al., 2018*). Upregulation was evident from an intermediate time point (9 days post RNAi), whereas epidermal genes downregulation occurred earlier (6 days post RNAi; 30%, 20/67; FDR<1E-5).

We hypothesized that biologically meaningful upregulation of neuron- and protonephridia-associated genes in neoblasts following *zfp-1* inhibition would result in overexpression of genes associated with these cell types in whole tissues, reflecting their increased production. To test this hypothesis, we examined published gene expression data from whole and regenerated tissues following *zfp-1* inhibition (*van Wolfswinkel et al., 2014*; *Zeng et al., 2018*). Predictably, the majority of downregulated genes were associated with the epidermis (70%, 490/698; Fold change < –2; FDR<1E-5; *Supplementary file 3*; Methods). Examination of upregulated genes showed that only a small fraction of the genes was associated with either protonephridial (4.9%) or neural (12.7%) expression at any examined time point (*Figure 5C*; *Supplementary file 4*; Methods). Therefore, gene expression analysis of whole tissues provided no evidence of an overrepresentation of neural or protonephridia cells in *zfp-1* (RNAi) animals, suggesting that cells related to these lineages were not excessively produced. Conversely, 37% of the upregulated genes following *zfp-1* (RNAi) in the whole tissue libraries were associated with intestinal cell types, despite no evidence for overrepresentation of intestine-associated FSTFs in neoblasts isolated from *zfp-1* (RNAi) animals compared to controls (*Supplementary file 4*).

In *zfp-1* (RNAi) neoblast colonies, we observed an absolute reduction in cycling cells (*Figure 3D*). We tested whether this phenotype was recapitulated in homeostatic, unirradiated, *zfp-1* (RNAi) animals by EdU labeling. We injected *zfp-1* or control double-stranded (dsRNA; Methods), and following 3 and 5 days, we performed EdU labeling by a 16 hr EdU pulse (Methods). Interestingly, there was an increase in EdU$^+$ nuclei in the *zfp-1* (RNAi) animals at both time points (*Figure 5D*). Investigating the potential correlation with apoptosis via TUNEL labeling revealed no early increase in cell death, which only increased at later stages (*Figure 5E*; Methods). Therefore, *zfp-1* inhibition does not trigger an acute rise in neoblast cell death that might induce hyperproliferation.

Colony sizes following *zfp-1* (RNAi) were reduced, yet cycling cell numbers in homeostasis were increased. Similar results were reported for the epidermal regulator *Smed-p53* (*Cheng et al., 2018*; *Pearson and Sánchez Alvarado, 2010*): neoblast colonies following *Smed-p53* (RNAi) are strikingly smaller (*Wagner et al., 2012*), yet in homeostasis, *Smed-p53* (RNAi) animals show an increased number of mitoses (*Pearson and Sánchez Alvarado, 2010*), and even an initial increase in canonical neoblast markers (*Pearson and Sánchez Alvarado, 2010*).

Considering that in homeostasis *zfp-1* (RNAi) planarians show greater numbers of cycling cells (*Figure 5D*) and intestinal gene upregulation in tissue (*Supplementary file 4*) as compared with controls, we conducted a targeted analysis to quantify the number of newly generated intestine progenitors in *zfp-1* (RNAi) animals. We counted intestine progenitors in situ (*Figure 5F*; Methods) and found a higher proportion of intestine progenitors in the *zfp-1* (RNAi) animals as compared with controls (*Figure 5F*; Mann-Whitney two-tailed U test p=0.008). Interestingly, in our analyses of growing colonies, we did not observe a parallel difference in intestine neoblast proportion (*Figure 4A*), leading us to speculate that the rise in cycling cells in homeostasis was an indirect effect of *zfp-1* suppression, and highlighting the limitations of neoblast analysis in colonies. In other words, integration of the analysis from homeostasis with our colony data suggests that *zfp-1* inhibition did not lead to a direct amplification of other lineages, but more likely to an indirect amplification of the intestine lineage.

## Discussion

Understanding how stem cell proliferation and lineage choices are regulated is critical for elucidating mechanisms of growth and homeostasis (*Chan et al., 2021*; *Dagan et al., 2022*; *Ferraro et al., 2010*; *Lei et al., 2016*; *Li and Xie, 2005*; *Morrison and Spradling, 2008*; *Scimone et al., 2010*; *Zhu et al., 2015*). Neoblasts form a simple yet powerful system, which has features of both adult stem cells and their transit-amplifying cell progeny (*Adler and Sánchez Alvarado, 2015*; *Baguñà, 2012*; *Reddien, 2018*). Functionally, neoblasts can repopulate stem cell-depleted planarians, produce a diversity of cell types, and adjust their growth rate according to organism requirements and nutrient availability (*Hayashi et al., 2006*; *Salvetti et al., 2009*; *Steele and Lange, 1976*; *Wang et al., 2018*). Herein, we sought to test whether growth in a neoblast colony is consistent with a model that does not assume communication between neoblasts or memory of cell fate decisions, but instead relies on stochastic

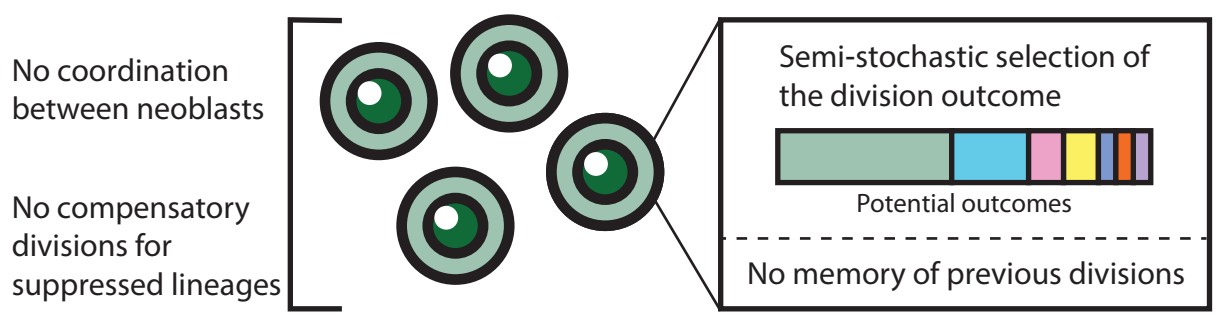

**Figure 6.** Model summarizing neoblast colony growth principles.

selection of division outcome (**Figure 6**). We further sought to enrich this model by understanding the interplay between neoblast lineage decisions and colony growth.

## Analysis of neoblast colony expansion

The initial model we developed was grounded in simple principles of colony growth (**Equation 1**, **Figure 2B**), in which cell fate decisions are made stochastically and independently of one another, as a function of the following parameters: average cell cycle length, symmetric renewal rate, and symmetric differentiation and neoblast elimination rate. We experimentally derived parameter values and compared the model's predictions to experimental data. Our findings suggested that the reliance on minimal assumptions provides a good description of neoblast colony growth following subtotal irradiation. The model captured the average colony growth within the tested time range, despite notable variability in observed colony sizes at a given time point. Based on these results, we suggest that if slow cycling neoblasts (**Molinaro et al., 2021**) emerge in colonies, their contribution to colony growth is likely negligible, allowing their exclusion from the model without sacrificing accuracy.

To better understand the potential consequences of random neoblast loss or excessive symmetric differentiation at early stages of colony growth, we used colony growth simulations, where each cell selected independently the outcome of its division. These simulations predicted that a fraction of colonies would fail to develop because of these random effects, a prediction that was corroborated by experimental results. This alignment demonstrates the utility of neoblasts as a system for studying stem cell colony formation and exploring concepts of stem cell behavior.

## Consequences of lineage-specific differentiation inhibition

Moving from the fundamental aspects of neoblast colony growth, we next tested how limiting lineage identity selection affected neoblasts. We suppressed the expression of key fate-specifying genes to dissect how stem cells respond to inhibition of their differentiation trajectory. The transition from measuring general colony growth to specific lineage perturbations allowed us to examine the association of proliferation and lineage specification. In particular, suppressing the master epidermal regulator *zfp-1*, corresponding to the most abundant planarian cell lineage (**Cheng et al., 2018**; **Raz et al., 2021**; **van Wolfswinkel et al., 2014**), drastically reduced neoblast colony growth. Interestingly, suppression of *Smed-p53*, a TF expressed in neoblasts and required for epidermal cell production, has resulted in a similar reduction in colony size (**Wagner et al., 2012**). These findings led us to consider how stem cells in a colony balance self-renewal and differentiation, and why both processes were disrupted by blocking differentiation to a single lineage. We suggest several interpretations for this observation: First, if we assume that a neoblast can alternate between self-renewal and production of differentiated progeny in each cell cycle, then a decision to produce a blocked progeny may preclude further divisions of the neoblast. A second possibility is that the presence of a specific lineage, epidermal progenitors in this case, is required for other neoblasts to proliferate; in this case, suppression of differentiation could indirectly limit colony growth. The first interpretation is more likely relevant for understanding colony growth given our observation that combined suppression of several smaller lineages produced a similar inhibition to colony growth and establishment.

The progressive reduction in colony size indicated that inhibition of a specific lineage reduced the fraction of neoblasts that contributed to colony growth, possibly due to their failure to produce

the blocked lineage and to alter their progeny identity. Interestingly, in unirradiated animals with abundant neoblasts, *zfp-1* inhibition led to increased cell proliferation – a response also noted after suppressing other regulatory genes like *Smed-p53* (*Pearson and Sánchez Alvarado, 2010*). This proliferation increase appeared before a rise in cell death, suggesting that cell death did not induce excessive proliferation in early stages. The difference in the consequence of lineage suppression in a neoblast colony and in unirradiated animals indicates that differential activity of feedback mechanisms regulates neoblast proliferation. We speculate that following subtotal irradiation, the near-complete depletion of neoblasts and their progeny (*Eisenhoffer et al., 2008*) alters the regulation of surviving neoblasts and leads to their rapid proliferation, which is not typical for neoblasts in homeostasis (*Wenemoser and Reddien, 2010*). Further investigation is needed into compensatory mechanisms that planarians use to maintain overall tissue integrity when differentiation processes are disrupted.

## A stochastic model for independent selection of cell division outcome in colonies

Our study investigates the mechanism behind neoblast division outcome selection. An exponential growth function describes the increase in colony size and the differentiation of progeny in the timescale of the analysis, closely aligning with our experimental results and published data (*Raz et al., 2021*; *Wagner et al., 2012*). Notably, this function operates without parameters for coordination of division outcome between neoblasts and neoblast memory. It posits that division outcomes are guided by an innate distribution of cell fates encoded within the neoblasts. We hypothesize that this distribution is modifiable by regional signals (e.g. Wnt, BMP), which indicate the neoblast location (*Gaviño and Reddien, 2011*; *Petersen and Reddien, 2008*; *Wurtzel et al., 2017*), and by injury signals (e.g. ERK), that modulate division rate (*Bohr et al., 2021*; *Fan et al., 2023*; *Tasaki et al., 2011*; *Wurtzel et al., 2015*). This strategy facilitates the dynamic balance in cell lineage production and meeting the organism's needs without direct communication between neoblasts and promotes return to homeostasis.

The emerging model enhances the understanding of neoblast dynamics and contributes to broader knowledge of stem cell behavior. The interplay between lineage-specific differentiation, proliferation, and resultant physiological adaptations presents a robust strategy for using pluripotent stem cells to re-achieve balance, in a manner that is not found in many non-regenerative organisms. This could inspire future research on pluripotent stem cells and their applications in regenerative biology.

## Methods
### Gene cloning and transformation
Selected genes were amplified using planarian cDNA (*Schmidtea mediterranea*, asexual isolate) and gene-specific primers and cloned into pGEM-T Easy vector using pGEM-T Easy Vector System I (Promega; CAT A1360). Vectors were transformed into *Escherichia coli* using the heat-shock method. Briefly, 5 µL of cloned plasmid were mixed with 100 µL of *E. coli* TOP10 bacteria and incubated on ice for 30 min. Next, bacteria were incubated for 45 s at 42°C, moved immediately to ice, and recovered in 350 µL of Luria Broth (LB) medium for 1 hr at 37°C. Then, 100 µL of recovered bacteria were plated on agarose plates containing 1:1000 ampicillin, 1:200 isopropyl β-d-1-thiogalactopyranoside (IPTG), and 1:625 5-bromo-4-chloro-3-indolyl-β-D-galactopyranoside (X-Gal). Plates were incubated overnight at 37°C, and colonies were screened by colony PCR using M13F and M13R primers with the following PCR program: (i) 5 min at 95°C; (ii) 34 cycles of 45 s at 95°C, 60 s at 55°C, and 2 min at 72°C; (iii) 7 min at 72°C; (iv) hold at 10°C. Reactions were analyzed by gel electrophoresis, and colonies having the correct fragment were grown overnight in LB medium, supplemented with 1:1000 ampicillin at 37°C at 180 rpm. Plasmids were purified with the NucleoSpin Plasmid Miniprep Kit (CAT 740588, Macherey-Nagel) and sequenced by Sanger sequencing. Primer sequences used for cloning the following genes: *hnf-4*, forward primer GATCTCGCACAATGCACTCG, reverse primer GTCTCACGAACTCCTTGCCA; *nkx2.2*, forward primer TTTGGTGCCAGCAGACTCAA, reverse primer TAGAGCCAGCTAATGTGGCG; *gata4/5/6*, forward primer CGGTATTGTCGAATTCTCACCAG, reverse primer TGACATCGCAATTGGAACCG; *foxF-1*, forward primer GTCCTATTTCCAGCACACAGC, reverse primer TCCGGAATCGTGCTGAGG.

## Near-complete neoblast ablation by subtotal irradiation

Animals were irradiated using a BIOBEAM GM 8000 (Gamma-Service Medical GmbH). To generate 1–3 colonies per animal, 2 mm starved (>7 days) worms were irradiated with 1750 rads (*Wagner et al., 2012*). Worms were allowed to recover and washed in planarian water a day after the irradiation.

## Synthesis of dsRNA for feedings and microinjections

DsRNA was synthesized as previously described (*Rouhana et al., 2013*). Briefly, templates for in vitro transcription were prepared by PCR amplification of cloned target genes using forward and reverse primers with flanking T7 promoter sequences on the 5' end. dsRNA was synthesized using the TranscriptAid T7 High Yield Transcription Kit (CAT K0441, Thermo Scientific). Reactions were incubated overnight at 37°C and then supplemented with RNase-free DNase for 30 min. RNA was purified by ethanol precipitation and resuspended in 70 µL of ultrapure $H_2O$. RNA was analyzed on 1% agarose gel and quantified by Qubit (CAT Q33223, Thermo Scientific) for validating a concentration higher than 5 µg/µL. Animals were starved for at least 7 days prior to RNAi experiments. In clonal expansion experiments, animals were fed with 14 µL of dsRNA mixed with 25 µL of beef liver 4 dpi. In homeostasis experiments, worms were injected four times with dsRNA every other day.

## Injections of dsRNA into planarians

Animals were injected with *zfp-1* dsRNA using Nanoject III (CAT 3-000-207, Drummond Scientific Company). Briefly, planarians were placed on a cold wet filter paper on their dorsal side and were injected posterior to the pharyngeal cavity. After the initial puncture, three consecutive dsRNA injections of 33 nL each were delivered at a rate of 66 nL/s. Worms were injected four times, every other day, and fixed at different time points for whole-mount analysis.

## Planarian fixation for whole-mount assays

Fixation was performed as previously described (*King and Newmark, 2013*). Animals were killed with 5% *N*-acetyl-L-cysteine (NAC, CAT 1124220100, Mercury) in PBS for 5 min, then incubated with 4% formaldehyde (FA) in 0.3% PBSTx (phosphate buffered saline, 0.3% Triton X-100) for 20 min. Animals were then washed in PBSTx, 50:50 PBSTx:methanol, and stored in methanol at –20°C.

## FISH using tyramide signal amplification

FISH was performed as previously described (*King and Newmark, 2013*) with minor changes. Briefly, fixed animals were bleached and treated with proteinase K (2 µg/mL, CAT 25530-049; Invitrogen) in PBSTx (phosphate buffered saline, 0.3% Triton X-100). Samples were incubated for 2 hr in a pre-hybridization buffer (pre-hyb) followed by an overnight incubation with probes. Samples were washed twice for 30 min in each solution: pre-hyb solution, 1:1 pre-hyb:2×SSCx, 2×SSCx, 0.2×SSCx, PBSTx. Blocking was performed in 0.5% Roche Western Blocking Reagent (CAT 11921673001; Sigma-Aldrich) and 5% heat-inactivated horse serum (CAT 04-124-1A; Biological Industries) in TNTx (100 mM Tris pH 7.5, 150 mM NaCl, 0.3% Triton X-100) for 2 hr. Animals were incubated with an anti-DIG-POD antibody (1:1500; Roche) or anti-DNP-HRP (1:10,000; Perkin-Elmer) overnight at 4°C. After antibody washes, tyramide development was performed as previously described (*King and Newmark, 2018*). Following development, peroxidase activation was quenched in a 1% sodium azide solution for 1 hr, followed by six PBSTx washes and antibody labeling for the second probe. Samples were labeled with DAPI (1:5000 in PBSTx) overnight at 4°C and mounted with Vectashield (CAT H-1000-10; Vector Laboratories).

## IF combined with FISH

Animals were fixed with 5% NAC, bleached, and treated with proteinase K as described for FISH analysis. Samples were incubated for 2 hr in pre-hyb followed by overnight incubation with the probes. Samples were washed twice in each solution, for 30 min each: pre-hyb solution, 1:1 pre-hyb:2×SSCx, 2×SSCx, 0.2×SSCx, PBSTx. Subsequently, blocking was performed in PBSTB (PBSTx, 0.25% BSA) for 2 hr at room temperature. Animals were then incubated with anti-SMEDWI-1 antibody (a gift from Peter W Reddien, 1:1000) at 4°C overnight. Then, seven PBSTx washes were performed, followed by incubation with PBSTB blocking solution for 2 hr at room temperature, followed by incubation with secondary antibody (goat anti-rabbit-HRP, 1:300 in PBSTB) overnight at 4°C. Post-antibody washes

and tyramide development (fluorescein tyramide; 1:2000) were performed as previously described (*King and Newmark, 2013*). After development, peroxidase activity was quenched using 1% sodium azide for 1 hr, followed by six PBSTx washes and incubation with anti-DIG antibody (1:1500; CAT 11207733910; Sigma-Aldrich) overnight at 4°C. Post-antibody washes and development with rhodamine tyramide (1:1000) were performed as described above. Samples were incubated with DAPI (1:5000 in PBSTx) overnight and mounted with Vectashield.

## FISH by hybridization chain reaction

Probe sets (30 pairs per gene) were designed and synthesized by the manufacturer (Molecular Instruments, Los Angeles, CA, USA) for the following genes: *tgs-1* and *smedwi-1*. Worms were fixed and bleached as previously described for FISH. Next, hybridization chain reaction (HCR) was performed according to the manufacturer's HCR RNA-FISH protocol for samples in solution (*Choi et al., 2018*). DAPI was added during the amplifier's wash steps, for a total incubation time of 2 hr at room temperature. Samples were mounted with Vectashield and stored at 4°C for subsequent analysis.

## Metabolic labeling by F-ara-EdU

F-ara-EdU (CAT T511293; Sigma-Aldrich) was first diluted in DMSO to the concentration of 200 mg/mL. Animals were soaked with 2.5 mg/mL F-ara-EdU diluted in planarian water for 16 hr, 24 hr after the indicated injections. For clonal expansion analysis, samples were soaked with EdU for 16 hr at day 11 post-irradiation. Samples were fixed, bleached, and treated with proteinase K as described for FISH (*King and Newmark, 2013*). Next, three washes in 3% PBSB (PBS supplemented with 3% BSA) were performed, followed by a click reaction using the baseclick kit (CAT Back-edu488, baseclick GmbH). Samples were washed three times with PBSB, incubated with DAPI (1:5000 in PBSTx) overnight at 4°C and mounted with Vectashield.

## IF labeling by anti-H3P labeling

H3P labeling was performed as previously described (*LoCascio et al., 2017*; *Wenemoser and Reddien, 2010*) with minor modifications. Briefly, following fixation with 5% NAC, bleaching, and proteinase K treatment, blocking was performed with 10% heat-inactivated horse serum for 2 hr. Next, anti-phospho-Histone H3 Antibody (CAT 04817; Sigma-Aldrich) was added in the concentration of 1:100 overnight at 4°C, followed by seven PBSTx washes. Samples were incubated in blocking solution for 2 hr and then incubated with goat anti-rabbit-HRP secondary antibody (Abcam; ab6721; 1:300) overnight at 4°C. Samples were washed seven times with PBSTx, developed using rhodamine tyramide diluted 1:1000 in PBSTi (PBSTx, 0.07% imidazole), labeled with DAPI overnight at 4°C, and mounted with Vectashield.

## Image acquisition and cell counting

Images of samples labeled by FISH, HCR, IF, F-ara-EdU, and TUNEL were collected using a Zeiss LSM800 confocal microscope. Labeled cells were counted manually using the cell counter module in the ImageJ software (*Rueden et al., 2017*). Normalization of EdU+ nuclei was performed by dividing the number of EdU+ nuclei by the average number of *smedwi-1*+ cells detected in a colony at the corresponding time point and divided by 10 for readability.

## TUNEL labeling

TUNEL was performed as previously described (*Pellettieri et al., 2010*) with minor modifications. Briefly, animals were fixed with 5% NAC and bleached overnight with 6% hydrogen peroxide in PBSTx. Animals were then treated with 2 µg/mL proteinase K for 10 min, 4% FA for 10 min, and were then washed twice with PBS. Using the ApopTag Red In Situ Apoptosis Detection Kit (CAT S7165; Sigma-Aldrich), five animals per tube were incubated with TdT enzyme mix for 4 hr at 37°C, followed by four PBSTx washes. Samples were incubated in a blocking solution (0.5% Roche Western Blocking Reagent and 5% inactivated horse serum in TNTx) for 2 hr followed by incubation with anti-DIG antibody (1:1000) overnight at 4°C. Samples were washed seven times with PBSTx, developed using rhodamine tyramide diluted 1:1000, labeled with DAPI (1:5000 in PBSTx) overnight at 4°C, and mounted with Vectashield.

## Assumptions underlying neoblast colony growth model

The model assumes that stem cell colony growth dynamics are governed by three key parameters: symmetric renewal probability, elimination probability, and cell cycle duration. Cell cycle duration represents the average time required for a single cell division. The model assumes universal probabilities for stem cells in the colony and does not incorporate environmental, niche, or intercellular effects. Additionally, no memory of previous cell divisions is assumed. While Raz et al. reported the potential of minor memory effects in neoblast divisions, these have been determined to have a negligible impact on the colony (*Raz et al., 2021*) and, therefore, on the model predictions. Symmetric renewal and elimination probabilities remain constant throughout the simulation, and fate specification is guided by predefined fate distributions derived from prior studies (*Raz et al., 2021*; *Scimone et al., 2014b*; *Scimone et al., 2018*; *van Wolfswinkel et al., 2014*).

## Estimation of the fraction of symmetric renewal divisions

Estimation of the fraction of symmetric renewal divisions was performed as previously described (*Lei et al., 2016*; *Raz et al., 2021*). Colonies at 7 dpi were labeled by FISH with the pan-neoblast marker *smedwi-1*. Colonies were closely examined using a confocal microscope (Zeiss LSM800) to find two adjacent *smedwi-1*$^+$ cells. Two spatially distinct labeled cells with high expression of *smedwi-1* marker were classified as symmetric division. Pairs composed of high *smedwi-1*$^+$ cells and low *smedwi-1*$^+$ cells were classified as asymmetrically dividing.

## Data extraction from published literature

Data was extracted using WebPlotDigitizer by labeling data points (*Drevon et al., 2017*). *Supplementary file 5* summarizes the extracted data and the original data source.

## Estimation of values of colony growth parameters

Counts of neoblasts in colonies at different time points following subtotal irradiation (*Figure 2F*; *Supplementary file 5*) were collected. Curve fitting using an exponential function (*Equation 2*) was performed with the curve_fit function from the scipy.optimize Python module using the average cell cycle length ($\tau$) extracted from the BrdU$^+$ labeling assay (*Figure 2C*, *Figure 2—figure supplement 1B–E*; *Lei et al., 2016*). The values of symmetric renewal ($p$) and symmetric differentiation or elimination ($q$) were estimated by fitting a range of values for $p – q$ between 0.1 and 0.4 and calculating the R-squared values of the fit to the data (*Figure 2—figure supplement 1D and E*). Experimental assessment of symmetric renewal ($p$) and asymmetric division ($a$) determined that they were equal (i.e. $p=a$; *Figure 2D*). The best fit for $p – q$, as estimated by examining the sum square of residuals (*Figure 2—figure supplement 1B–E*), was found to be 0.345. Since (1) all possible division outcomes are $p+q+a = 1$; (2) $p=a$; and (3) $p – q = 0.345$, we estimated that $p = 0.45$ and $q = 0.1$.

## Simulation of colony growth

Simulations were performed using custom R code, executing as follows: selection of parameters occurred using selected frequencies of symmetric renewal and symmetric differentiation or elimination. Initial colony size was determined by randomly selecting a value from the empirical distribution of early colony size, as reported in the literature (*Raz et al., 2021*; *Wagner et al., 2011*). In each simulation round, every cell underwent selection of an outcome, adhering to the predefined frequencies. This simulation process extended 5–20 cycles and was replicated 100 times. Code simulating empirical colony growth and cell fate choice is available at GitHub (copy archived at *Frankovits, 2025*).

## Gene expression analysis of S/G2/M neoblasts

Processed differential gene expression data profiling FACS-purified S/G2/M neoblasts isolated from control and *zfp-1* (RNAi) animals was downloaded from PLANAtools (*Hoffman and Wurtzel, 2023*). Each gene in the table was assigned a cell-type identity based on the planarian cell-type gene expression atlas (*Fincher et al., 2018*).

## Statistical analyses

Statistical tests were performed using GraphPad Prism (v 10.0.3), using the scipy stat module (v 1.12) and R (v 4.3.2). Statistical significance was assessed using the Mann-Whitney two-tailed U test or

two-tailed Student's t-test, and the threshold for considering an effect significant was 0.05, unless stated otherwise.

## Acknowledgements

We thank Prof. David Sprinzak, Prof. Iftach Nachman, and Dr. Yasmine Meroz for their critical input. We thank Shrey Jain for his assistance with performing FISH. We thank members of the Wurtzel lab for feedback and discussions. We thank Dr. Daria Makarovsky from the interdepartmental services at Tel Aviv University's Faculty of Medicine for assistance with gamma irradiation. OW is supported by the Israel Science Foundation (grant 2039/18) and the European Research Council (no. 853640). OW is a Zuckerman Faculty Scholar.

## Additional information

### Funding

| Funder | Grant reference number | Author |
| --- | --- | --- |
| European Research Council | 10.3030/853640 | Prakash Varkey Cherian<br>Yarden Yesharim<br>Omri Wurtzel |
| Israel Science Foundation | 2038/18 | Tamar Frankovits<br>Yarden Yesharim |

The funders had no role in study design, data collection and interpretation, or the decision to submit the work for publication.

### Author contributions

Tamar Frankovits, Conceptualization, Data curation, Validation, Investigation, Visualization, Methodology, Writing – original draft, Writing – review and editing; Prakash Varkey Cherian, Yarden Yesharim, Investigation, Visualization; Simon Dobler, Investigation; Omri Wurtzel, Conceptualization, Resources, Data curation, Software, Formal analysis, Supervision, Funding acquisition, Validation, Investigation, Visualization, Methodology, Writing – original draft, Project administration, Writing – review and editing

### Author ORCIDs

Tamar Frankovits ⓘ https://orcid.org/0009-0002-1462-3017
Omri Wurtzel ⓘ https://orcid.org/0000-0002-6865-5914

Reviewer #1 (Public review): https://doi.org/10.7554/eLife.100885.3.sa1
Reviewer #2 (Public review): https://doi.org/10.7554/eLife.100885.3.sa2
Author response https://doi.org/10.7554/eLife.100885.3.sa3

## Additional files

### Supplementary files

Supplementary file 1. Downregulated genes in G2/M neoblasts following zfp-1 (RNAi).

Supplementary file 2. Upregulated genes in G2/M neoblasts following zfp-1 (RNAi).

Supplementary file 3. Downregulated genes in tissues or blastema following zfp-1 (RNAi).

Supplementary file 4. Upregulated genes in tissues or blastema following zfp-1 (RNAi).

Supplementary file 5. Summary of data extraction using WebPlotDigitizer.

MDAR checklist

### Data availability

Data generated or analysed during this study are included in the manuscript and supporting files. *Figure 2C and E*, source data in *Supplementary file 5*. *Figure 5*, source data in *Supplementary*

*files 1–4*. Code simulating empirical colony growth and cell fate choice is available at GitHub (copy archived at *Frankovits, 2025*).

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
