## [Editor Report · eLife Assessment]

This manuscript establishes a mathematical model to estimate the key parameters that control the repopulation of planarian stem cells after sublethal irradiation as they undergo fate-switching as part of their differentiation and self-renewal process. The findings are **important** for future investigation of stem cell division in planarians and have implications for analyzing stem cell biology in other systems. The methods are **convincing**, integrating modeling with perturbations of key transcription factors known to be critical for cell fate decisions, but the authors have only shown that this is the case for a small number of stem cell types.

---

## [Referee Report · Reviewer #1 (Public review)]

Summary:

This is a very creative study using modeling and measurement of neoblast dynamics to gain insight into the mechanism that allows these highly potent cells to undergo fate-switching as part of their differentiation and self-renewal process. The authors estimate growth equation parameters for expanding neoblast clones based on new and prior experimental observations. These results indicate neoblast likely undergo much more symmetric self-amplifying division than loss of the population through symmetric differentiation, in the case of clone expansion assays after sublethal irradiation. Neoblasts take on multiple distinct transcriptional fates related to their terminally differentiated cell types, and prior work indicated neoblasts have a high plasticity to switch fates in way linked to cell cycle progression and possibly through a random process. Here, the authors explore the impact of inhibition of key transcription factors defining such states (ie "fate specifying transcription factors", FSTFs) plus measurement and modeling in the clone expansion assay, to find that inhibition of factors like zfp1 likely cause otherwise zfp1-fated neoblasts to fail to proliferate and differentiation, without causing compensatory gains in other lineages. A mathematical model of this process assuming that neoblasts do not retain a memory of prior states while they proliferate and transition across specified states can mimic the experimentally determined decreased sizes of clones following inhibition of zfp1. Complementary approaches to inhibit more than one lineage (muscle plus intestine) supports the idea that this is a more general process in planarian stem cells. These results provide an important advance for understanding the fate-switching process and its relationship to neoblast growth.

Overall I find the evidence very well presented and the study compelling, and offers an important new perspective on the key properties of neoblasts. I have some comments to clarify the presentation and significance of the work.

Comments on revisions:

In this revised version, the authors nicely address all of my comments and I find the work makes a strong case for its main conclusions.

---

## [Referee Report · Reviewer #2 (Public review)]

Summary:

Cell cycle duration and cell fate choice are critical to understanding the cellular plasticity of neoblasts in planarians. In this study, Tamar et al. integrated experimental and computational approaches to simulate a model for neoblast behaviors during colony expansion.

Strengths:

The finding that "arresting differentiation into specific lineages disrupts neoblast proliferative capacities without inducing compensatory expression of other lineages" is particularly intriguing. This concept could inspire further studies on pluripotent stem cells and their application for regenerative biology.

Comments on revisions:

The authors have addressed all of my comments and concerns.

---

## [Author Response]

The following is the authors’ response to the original reviews.

**Public reviews**

**Reviewer #1 (Public review):**
Overall I find the evidence very well presented and the study compelling. It offers an important new perspective on the key properties of neoblasts. I do have some comments to clarify the presentation and significance of the work.

We thank the reviewer for the positive feedback and plan to improve the presentation of the work.

**Reviewer #2 (Public review):**
However, the absence of a cell-cell feedback mechanism during colony growth and the likelihood of the difference needs to be clarified. Is there any difference in interpreting the results if this mechanism is considered?We will improve the description of the model assumptions and the interpretation of the data on the basis of these assumptions.Although hnf-4 and foxF have been silenced together to validate the model, a deeper understanding of the tgs-1+ cell type and the non-significant reduction of tgs-1+ neoblasts in zfp-1 RNAi colonies is necessary, considering a high neural lineage frequency.

We will improve the analysis of this result in light of the experimentally determined frequency of the *tgs-1+* neoblast population.

**Recommendations for the authors**

**Reviewing Editor Comments:**
After consultation, we have compiled a list of the key changes to be made to the manuscript, along with reviewer-specific recommendations to follow.(1) Include a section that explicitly describes the assumptions and limitations of the study, particularly with respect to the following assumptions:

We thank the reviewers for the comment. We added a description of the model assumptions in the methods section “Assumptions underlying neoblast colony growth model”.

a) All known types of specialized neoblasts cycle at the same rate (see points from Reviewer 1).

We thank the reviewers for the comment. The current data used to estimate τ (Lei et al., *Dev Cell*, 2016) does not allow the direct estimation of individual cycling behaviors. Consequently, we assume that all specialized neoblasts cycle at the same average rate, a simplification supported by the model's accurate prediction of colony growth.

b) The assumption that any FSTF-like gene would behave like zfp1 or foxF and hnfA genes. The manuscript does not mention that there may be fundamental differences among these different FSTFs that could be uncovered by future work. A strong addition to the paper would be to test other epithelial genes (e.g. p53, chd4, egr5) to show reproducible behavior within a single lineage.

We thank the reviewers for the comment. Colony size reduction following inhibition of *Smed-p53* and failure to produce epidermal progenitors is strongly supported by previous analysis (Wagner et al., *Cell Stem Cell, 2012*). We refer to this observation in the paper in the section titled: “Inhibition of *zfp-1* does not induce overexpression of other lineages in homeostasis”. We added the following sentence to the discussion (Line 460-462): Interestingly, suppression of *Smed-p53*, a TF expressed in neoblasts and required for epidermal cell production, has resulted in a similar reduction in colony size (Wagner et al., *Cell Stem Cell,* 2012).

Of note, *Chd4* expression is not limited to specialized neoblasts or to a specific lineage (Scinome et al., *Development*, 2010), and therefore its inhibition likely has a more complex outcome than an effect on a single lineage. Furthermore, *egr-5* is not expressed in neoblasts (Tu et al, *eLife*, 2015), making this experimental condition more challenging to examine in the context of neoblast colonies at the time points assessed in this study.

c) The fact that the data used to feed the model relies on radiated animals which are likely to have altered cell cycle rates compared to unirradiated animals (see comment by Reviewer 1). Of note, the model predicts a steady increase in colony size, but colony size does not change between 9dpi and 12dpi.

We thank the reviewers for the comment. The colony size in control animals increased between 9 and 12 dpi (Fig 3B), as predicted by the model. In *zfp-1* (RNAi) animals, the median colony size has also increased over this period, at a slower rate, which we attribute to the increase in *q*. We attribute the unchanged average colony size to an increase in the frequency of cells failing to proliferate, because of selection of a fate they cannot fully differentiate into.

d) In light of both reviewers' comments about colony expansion vs. feedback, the authors should discuss how predicted changes to division frequencies might change as homeostasis is reached, or explain how their model accounts for the predicted rate differences under homeostatic conditions in which overall neoblast numbers do not change. Can the model estimate when this transition might occur?

We thank the reviewers for the comment. Our colony assays are constrained by the animals survival following sub-total irradiation (16 to 20 days). In this timeframe, the neoblast population is overwhelmingly smaller in comparison to non-irradiated animals. Therefore, the animals do not reach homeostasis during the experiment, and the model does not allow to estimate the time the system would need to return to homeostasis.

(2) In Figure 2D, the assumption is that these adjacent smedwi-1+ cells are sisters. Previous data analyzing this relied on EdU or H3P staining to show a shared division history. When these images were collected is therefore extremely critical to include (the methods suggest 7, 9, or 12 days). The authors should justify why they believe that these adjacent cells are derived from a single neoblast that has divided only once.

We thank the reviewers for the comment. The images were collected at 7 dpi. We modified the figure legend and the associated methods to include this information. At this early time point, *smedwi-1*+ cell dyads are spatially separated from other neighboring cells, suggesting that they are the product of a single cell division. Importantly, our data is in complete agreement with previous estimates of symmetric renewal division rate (Raz et al., *Cell Stem Cell*, 2021; Lei et al, *Developmental Cell*, 2016).

(3) Clarify the wording 'pre-selected' in the abstract as described by Reviewer 1.

We thank the reviewers for the comment, and for clarity we replaced the wording “pre-select” with “select”.

(4) Experimental details that are important to the interpretation should be added. For example, how is belonging to a colony defined? This is important because some of the data (e.g. Figure S1A: similar numbers of smedwi-1+ cells are observed at 2dpi and 4dpi, but 4dpi is considered a colony whereas 2dpi is not). The timing of quantification should be included in each figure (it is missing in Figure S2, and Figure 3C and 3D). How the authors distinguish biological vs technical replicates is not mentioned.

We thank the reviewers for the comment. Subtotal irradiation may result in formation of a spatially-isolated cluster of neoblasts that is not distributed throughout the animal (Wagner et al., *Science*, 2011). This localized cluster of neoblasts is defined as a neoblast colony (Wagner et al., *Science*, 2011; Wagner et al., *Cell Stem Cell*, 2012). The small number of high *smedwi-1*+ cells observed at 4 dpi in our experiments aligns with this definition (Fig S1A). By contrast, the low *smedwi-1* expression detected across the animal 2 dpi does not fit this definition and likely reflects remnants of dying neoblasts resulting from irradiation. The following text was added to the figure legend: “isolated cells expressing low levels of *smedwi-1*+ were scattered in the planarian parenchyma, likely reflecting remnants of dying neoblasts”.

(5) Figure 5F appears to use SMEDWI-1 antibody (based on capital letters and increased signal in the brain). Is this the case? The methods do not mention the use of a SMEDWI-1 antibody, and the text indicates that these are progenitors, but SMEDWI-1 protein is well known to not mark neoblasts. If the antibody was used, the authors should not claim that these are neoblasts.

We thank the reviewers for the comment. The SMEDWI-1 antibody used in the experiments described in Figure 5F indeed labels neoblasts and their progeny (Guo et al., *Developmental cell*, 2006). The methods section “Immunofluorescence combined with FISH” details the labeling procedure, which combines FISH and IF using this antibody.

All microscopy images are difficult to see. Perhaps this is because they are formatted as CMYK images. They should be converted to RGB format to make them appear less dull.

We thank the reviewer for the comment. Improved version of the figures has now been uploaded.

The terminology used in Figure 5 to describe upregulation should not be "overexpression". We thank the reviewers for the comment.

We changed the terminology to “upregulated”.

**Reviewer #1 (Recommendations for the authors):**
I think the authors should include a section that explicitly lays out the assumptions and limitations of the study. For example, I believe that determining tau requires assuming that all different types of specialized neoblasts cycle at the same rates. Also there is the assumption that any FSTF-like gene would behave like zfp1 or foxF and hnfA genes. It seems to remain possible that a future study could find that a subset of FSTFs might indeed exert "either/or" decisions in fating, just not the particular genes under investigation here.

We thank the reviewer for the comment. We added a description of the model assumptions in the methods section.

In the abstract, the wording "pre-selected" is somewhat puzzling to me. I would interpret a preselection as a process that defines the next specified state prior to its manifestation. Instead, and as I understand the authors argue this as well, the study provides good evidence that the determination mechanism is random in that subsequent neoblast choices do not likely depend on prior states. So I would suggest changing that wording.

We thank the reviewer for the comment. We replaced “pre-select” with “select”

Is it possible to determine the uncertainty in measuring tau the cell cycle time and would this have an impact on subsequent modeling?

We thank the reviewers for the comment. The current data that was used to estimate tau (Lei et al., *Dev Cell*, 2016) does not allow us to directly estimate the uncertainty in measuring τ.

For lines 154-164 I would suggest doing a little more to explicitly write out the logic of determining the growth constants within the main text and not just in methods, for ease of reading.

We thank the reviewer for the comment, and added explanations for how we determined the growth constant in the text. The text now reads (lines 160-166): “Considering an average cell cycle length of 29.7 hours, we calculated the value of *q* using the following approach: the probabilities of all cell division outcomes must sum to 1. Our experimental data showed that symmetric renewal (*p*) and asymmetric division (*a*) occur at equal rates (i.e., *p* = *a*). By fitting these parameters to the experimental data, we determined that the difference between the probabilities of symmetric renewal and symmetric differentiation (i.e., *p* - *q*) was = 0.345 (Fig 2E, S1D-E). Therefore, with these criteria, we estimated the probabilities of cell division outcomes in the colony as *p* = 0.45, *a* = 0.45, and *q* = 0.1 (Fig 2G; Methods).”

Line 192 why does post-mitotic progeny number linearly relate to neoblast number? In clones, a change in *q* has an exponential effect. I feel like I am missing something.

We thank the reviewer for the comment. In colonies, 50% of cell divisions result in the production of post-mitotic progeny (asymmetric division). Therefore, the number of produced progenitors in a given cell cycle is linearly correlated with the number of neoblasts. This statement is in line with previous analysis of planarian colony size (Wagner et al., *Cell Stem Cell*, 2012).

Line103 it also seems possible, although less likely, that the specified state is not fixed within a given cell cycle and could be that cells that try to switch into zeta-neoblasts mid-cell cycle arrest in proliferation etc just for that time.

We thank the reviewer for the comment and agree that this is a possibility. However, our observations suggest that incorporating this factor into the model is unnecessary for accurately predicting colony size.

In terms of the feedback mechanism proposed to operate in homeostasis, I think in the case of zfp-1 it is quite likely that loss of epidermal differentiation results in wound responses (this phenomenon has been documented in egr-5 RNAi in Tu et al 2015 I believe). This could play out differently in the clone assay because the effects of sublethal irradiation on this process would predominate in both control versus zfp1(RNAi) conditions.

We thank the reviewer for the comment. Our RNA-seq analysis following *zfp-1* inhibition did not show overexpression of injury-induced genes at an early time point (6 days; Fig. 5B-C). However, an increase in cycling cells was detected much earlier via EdU labeling (3 days; Fig. 5D). In the case of *egr-5* suppression, Tu et al. analyzed injury-induced gene expression at a later stage (21 days of RNAi), where they found significant epidermal defects (see Fig. 5C in Tu et al.). We agree that sublethal irradiation effects likely predominate in colony analysis for both control and *zfp-1* (RNAi) animals. In homeostasis, additional factors likely influence cell proliferation and differentiation.

It seems likely that some of the differences noted between homeostasis versus clone growth could ultimately arise from the different growth parameters under each setting. Could the rate parameters be estimated from prior data in homeostasis as well? It seems to me that with the framework the authors use, homeostasis must involve a net zero change to neoblast abundance (also shown by Wagner 2011 by the sigmoidal curve of neoblast abundance at the endpoint of clone expansion). Therefore, in these conditions p=q by definition. Experimental evidence from Lei 2016 (Figure S7M) suggests asymmetric divisions and symmetric renewing divisions are about equally abundant (5/12 41% sym renewing vs 7/12 69% asymmetric renewing). Therefore, under homeostasis, there would be an estimated p=q=0.3 and a=0.4. Compared to clone growth conditions then, in homeostasis, it seems that roughly the rate of symmetric renewal decreases and the rate of symmetric differentiation also increases. I wonder, could this kind of difference potentially account for the differences between homeostasis versus clone expansion settings? It is also worth noting that the clone expansion context has been used as a sensitized genetic background for identifying effects of gene inhibition on neoblast self-renewal, so perhaps the reason this works is that the rates of selfrenewal are relatively less in homeostasis so that clone expansion represents a case where there is greater demand for self-renewal.

We thank the reviewer for the comment. We agree that under homeostatic conditions, where the population size remains stable, the average probability of symmetric renewal matches the average probability of symmetric differentiation or elimination. By contrast, during colony expansion, the probability of symmetric renewal exceeds that of symmetric differentiation or elimination. The differences in response to a lineage block between homeostasis and colony expansion can have multiple interpretations. However, data from homeostatic animals does not permit the analysis of individual neoblasts or their specific responses to a lineage block. Consequently, we cannot determine whether the proliferative response following the lineage block during homeostasis is a direct response to the lineage block or an indirect effect resulting from changes in other neoblasts. We discuss these possibilities further in lines 472 - 484.

In terms of the memory effect, I recall some arguments presented in the Raz 2021 study that were consistent with a slight memory for neoblast specification being retained. I believe this was a minor point from detecting a slightly higher likelihood of identifying 2-cell clones that both took on prog1+ identity compared to the population average. If this is the case, it may be worth the authors commenting on reconciling those observations with their model.

We thank the reviewer for their comment. Raz et al. (*Cell Stem Cell*, 2021) reported that in the asymmetric division of a zeta-neoblast, which generates a *prog-2+* cell and a neoblast, there was a slightly higher observed frequency of *zfp-1* expression in the neoblast compared to the expected rate (Expected: 32%, Observed: 44%). This small increase may reflect a mild memory effect, experimental variability, or both. However, statistical analysis using Fisher's exact test yielded a non-significant p-value (p = 0.1), suggesting that this difference could be attributed to experimental variability. Other data from Raz et al., such as lineage representation in early colonies, also did not show significant memory effects, indicating that any such effects, if present, are minimal and difficult to detect. Therefore, while we do not, and cannot, rule out the presence of minor memory effects, we expect that effects of this magnitude will have minimal impact on our model.

**Reviewer #2 (Recommendations for the authors):**
Figure 2C and 2D:Please provide the specific time points for the data presented.

We thank the reviewer for the comment. The information was added to the figure legend.

Colony growth and homeostasis:It would be beneficial to estimate a time point at which colony growth transitions to a model with a cell-cell feedback mechanism, similar to that observed in homeostasis. This would help in understanding the dynamics and timing of these processes.

We thank the reviewers for the comment. Our colony assays were constrained by the animals survival following sub-total irradiation (16 to 20 days). Neoblast numbers are substantially reduced compared to unirradiated animals, preventing us from determining the time point at which homeostasis is achieved.

Methods:μl should be μL

The text was changed accordingly.

Line 526: H2O should be H2O

The text was changed accordingly.